# Viewing of abstract art follows a gist to survey gaze pattern over time regardless of broad categorical titles

Eugene McSorley[1]*, Rachel McCloy[1], Louis Williams[2]

**1** School of Psychology & Clinical Language Sciences, University of Reading, Harry Pitt Building, Earley Gate, United Kingdom, **2** ICMA Centre, Henley Business School, University of Reading, United Kingdom

* e.mcsorley@reading.ac.uk

## Abstract

Artworks are often shown alongside informative contextual text. This can be as simple as its title and date, or it can be a more descriptive or elaborative explanation of the piece, such as describing its content or explaining the symbolism of the work. Contextual information has been shown to have no impact on early viewing behaviour. A consistent impact on observer eye movement scan paths and fixations is only found in the later periods of looking. One explanation for this is that early eye movement responses to artworks support automatic low-level visual processing that quickly extracts a broad, holistic gist or sense of the works, allowing rapid categorization. The influence of contextual information may only be felt in later periods as it concentrates fixations on descriptive or elaborative elements more quickly and for longer periods. To examine this explanation, we recorded eye movement responses to abstract paintings that were preceded by basic-level categorical labels. In Experiment 1, abstract works were preceded by the labels Landscape, Portrait or Abstract, while in Experiment 2 the labels Action, Still Life or Abstract were given. We found that saccadic eye movements were more common in the first 2 seconds of viewing and were spread over a large portion of the artworks. They became less frequent and were spatially clustered over time. This changing pattern of saccades and their fixation periods was not affected by the contextual information provided by the category labels suggesting a minimal role for top-down control of eye movements when viewers are faced with abstract artworks.

## Introduction

Art tourism and museum visits are increasing as financial and social barriers decrease. As a result, the importance of contextualising museum collections becomes more important to help maintain and improve visitor experience, improve access, and deepen appreciation and understanding. For example, knowledge about

**Data availability statement:** All data files used to support this submission, including the submitted manuscript, figures and supplementary materials are publically available at Open Science (DOI 10.17605/OSF.IO/EGH25; https://osf.io/egh25/).

**Funding:** The author(s) received no specific funding for this work.

**Competing interests:** The authors have declared that no competing interests exist.

the symbols and allusions in art that may have been commonplace amongst visitors in past may not be able to be taken for granted anymore.

Artworks are usually contextualised through the titles they are given or supplemented with brief descriptive information that accompanies them. These can be chosen by the artist to reflect artistic intention [1–3] and they may be descriptive or explanatory, to draw attention to or revealing intentions in the works, or they may be designed to give no information, misinform, or misdirect viewers. Contextual information can sometimes be given, commonly in text written by curators. This may be specific to a work, or it may be more widely applicable to a whole show. How useful titles or contextual information is to visitors is debatable [4,5], although evidence does suggest that they impact visitor experience in museums, with viewers generally adopting an art-label-art behaviour pattern which occurs cyclically as the viewer continues to interact with the artwork [6–12].

The gaze patterns of viewers across artworks are initially typified by short fixations separated by long distances achieved through large amplitude saccadic eye movements which, over time, are replaced with periods of longer fixation on select areas of interest which are separated by shorter amplitude saccades [13–17]. This shifting pattern of gaze parameters has been linked to two types of processing: an ambient mode followed by focal mode processing [18]. Sometimes referred to as the gist phase and survey phase respectively [15,19–21]. These have been suggested to reflect an initial extraction of the gist of the artwork early in the viewing period, followed by a phase of viewing in which select areas are concentrated on by the viewer. This fits well with many of the current multi-stage models of aesthetic experience. Here global visual properties and features of the works are first extracted followed by the higher-level cognitive processing of those features, with further analysis of content along with art style, which is highly dependent on the viewer's expertise and experience, to achieve a level of viewer understanding before a judgement is made along with an aesthetic response [15,22–25]. Thus, initial rapid scanning of the artwork acts to support the extraction of the gist of the work and is guided by the parallel perceptual processing of holistic visual features of the work. This is gradually replaced with fixations driven by contextual information to support efforts for meaning-making, reflective processing, and cognitive understanding of the artwork [25].

Models such as these make the direct predictions that providing contextual information about artworks will improve meaning-making and understanding of the works and will impact the viewer's aesthetic appreciation and experience. To support this, viewer's gaze patterns when looking at art are affected by the presence of contextual information in the form of titles and descriptions. Fixation distribution across representational artworks is wider when contextual information about artworks is given, and time spent on areas of interest included in the contextual information is increased [26–29]. Furthermore, the distance travelled between fixations (saccade amplitudes) is shorter [28] and returns to areas of interest occur more frequently [27]. This has been found to differ depending on the style of painting and task undertaken [28–31]. For example, when presented with works by Kandinsky, viewer's gaze was found to concentrate on those areas indicated in the title (e.g., *Painting with white*

border) and to look at them more often [27]. This pattern of viewing was not found for those who were not made aware of the title. Likewise, changing the depth of information from simple, factual information (including the title, artist name and dates, date of painting and medium and technique) to elaborative descriptions (including the theological and symbolism) of Zurbarán's paintings of Jacob and his Twelve Sons, showed significant differences in viewing patterns [26]. Elaborative information evoked wider fixation distributions and more time spent on areas of symbolic importance. Consistent effects of titles on gaze behaviour are only reported during later viewing periods. For example, viewing behaviour in response to Kandinsky's paintings [27] and Zurbarán's paintings [26] showed no effect of title on early viewing periods. Title effects only revealed themselves later, after the first 2 seconds. Similarly, while viewing works by Dali (surrealist such as *Swans Reflecting Elephants*) and Caravaggio (baroque such as *The Sacrifice of Isaac*), [28,30] reported effects of title and task on viewing behaviour. The earliest effects varied depending on the artist style and whether the viewer was asked to give a preference or subsequently describe the work. This fits well with early verbal descriptions of artworks by artists such as Bruegel, Vermeer, and Klee [15], which, along with early eye movements, reflect their structural elements, semantic meaning and overall gist. Effects of contextual information only occur after the first 2–5 seconds of viewing [15,27,32]. After 7 seconds, verbal reactions begin to reflect higher-level concepts and comments on art styles, forms, and emotional reactions to the works [32].

The lack of consistent effect of titles on early viewing behaviour and verbal responses to artworks suggests that the reason why descriptive titles have been found not to influence early eye movement responses is because they simply describe those aspects of the artworks that are captured by their gist (i.e., the structure and semantic meaning). Being made aware of them before viewing an artwork does not affect scan paths as these are exactly the features which are already being used to guide those early eye movement responses. Elaborative titles which highlight information beyond these basic aspects of the artworks would only affect those eye movements which occur later in viewing as the focal survey phase comes into play. Given the quick gist extraction of artworks, we suggest that rather than giving descriptive or elaborative titles as prior contextual information (titles) a more effective contextual guide for viewing behaviour would be to give simple basic-level or superordinate categorical information. If categorical information can be used to guide expectations of what artworks depict then we would expect to see differences in eye movement control, especially early in the viewing period. We know that briefly presented commonly encountered heterogeneous scenes (10–107ms) are easily categorized at a basic-level [33–43]. Initial holistic impressions of artworks are also reported with very brief exposures and show stable ratings with increasing exposure durations based on style, structure, semantic meaning, complexity, harmony, and order [15,21,44–47]. There is some evidence that aesthetic judgements of beauty, liking, impressiveness, and specialness can be extracted with brief exposures, but other studies report that this develops over time [21,48,49].

To examine this explanation, we presented participants with a series of abstract paintings created in-house, which contained little in the way of semantic meaning or structure that could be captured by a descriptive title. Only prior information about the basic- or superordinate-level categorization of the artworks was given. As task requirements have been shown to change viewer's behaviour, we asked participants to look at the artworks as artworks only thereby encouraging them to adopt an aesthetic stance regarding each piece in the context of the preceding categorical information. They were not asked to judge the works for liking or understanding. In Experiment 1 the paintings were categorised and labelled as being an abstract work of a Landscape or a Portrait or were simply labelled Abstract. In Experiment 2 the pieces were labelled as being abstract attempts to capture Still life or Action or were again labelled Abstract. We hypothesized that in both experiments, eye movement patterns would show a progression from the gist-ambient mode of viewing, showing large amplitude saccades separated by short periods of fixation, to the more focal-survey mode of viewing, where patches of the abstract work will be focused on showing a pattern of small amplitude saccades and longer fixations. We further hypothesized that for Experiment 1, more horizontally oriented saccadic movements would be elicited for works labelled as Landscape, with more vertically oriented saccades for those labelled as Portraits both relative to each other and to

those labelled as Abstract. For Experiment 2, we hypothesized that those labelled as being works showing Still life would evoke fewer saccades with longer periods of fixation relative to those labelled as depictions of Action and as Abstract. While those labelled as Action would produce more saccades separated by shorter fixations with higher velocities than those labelled as Abstract works.

To pre-empt the results, we find that viewing behaviour of abstract paintings starts with a rapid scanning of the image before dropping into a pattern of shorter movements and longer fixation periods, showing a switch from ambient to focal processing style, that is not affected by category label. This suggests eye movements made in the first few seconds of exposure to abstract artworks are driven by the structural elements of the piece and are resistant to even the most basic of contextual labels.

## Methods

### Participants

Sample size was determined using G*Power 3.1 [50]. Number of groups was set at 1 and measurements at 9 (3X3 two-way repeated measures design), with alpha set at .05 and power at .8. The assumption that the correlation between the two levels of the repeated measures factors was set at $r = .5$ and a medium effect size of Cohen's $f = .25$ was selected. As a result, the total sample size was determined to be 15. This is in line with comparable studies examining the impact of art on eye movement behaviour (see studies examining eye movements in changes in response to art discussed in the introduction, which is typically around 20 overall or per condition: 14, 16, 20, 26–30). An over-recruitment strategy was adopted in cases of participant dropout and tracker loss or random factors such as unexpected fire alarms. Overall, 52 participants took part in both experiments, 30 in Experiment 1 and 22 in Experiment 2. Half of the participants were female. Age ranged between 19 and 55 years of age with an average of 23. Data was collected between 8th October 2019 and 1st March 2020 at the University of Reading, UK. All had normal or corrected to normal vision. Ethical approval from the Ethics Committee of the School of Psychology & Clinical Language Sciences, University of Reading, UK, was granted for this study. Informed consent for taking part was obtained from all participants.

### Apparatus & materials

In both experiments, the same 30 abstract paintings were used and were created in-house by two Final Year students studying for their joint honours Batchelor of Arts degree in Art & Psychology at the University of Reading. They were created using oil paints, inks and spray paints. The artworks were digitized, sized to 13 by 13 degrees of visual angle, and presented in color on a 21" Diamond Pro computer monitor with a refresh rate of 75 Hz at a viewing distance of 1m (See Fig 1 for examples and S1 for full stimulus set). An SR Research Eyelink II eye-tracker was used to track eye movements in cornea and pupil mode. This gave a sample rate of 250 Hz.

### Design

A repeated measures design was used in both experiments with participants viewing each of the 30 abstract works. In Experiment 1 they were labelled "Landscape", "Portrait" and "Abstract". In Experiment 2 they were labelled "Still Life", "Action" and "Abstract". Ten works were randomly assigned to these labels for each participant. Labels were randomly presented across the experiment and were not blocked by content type. They were presented prior to the artworks in Verdana font for 3 seconds. The paintings were then shown for 15 seconds in Experiment 1 and 10 seconds in Experiment 2. This change was made as participants in Experiment 1 commonly reported that 15 seconds was too long a viewing period given the abstract nature of the artworks. To examine changes in fixation and eye movement behavior across the viewing period we introduced a second factor of Time. Eye-tracking data was sectioned into that which occurred early (0–2 seconds), mid-way (2–7 seconds), and towards the end of the trial (7 seconds to trial end).

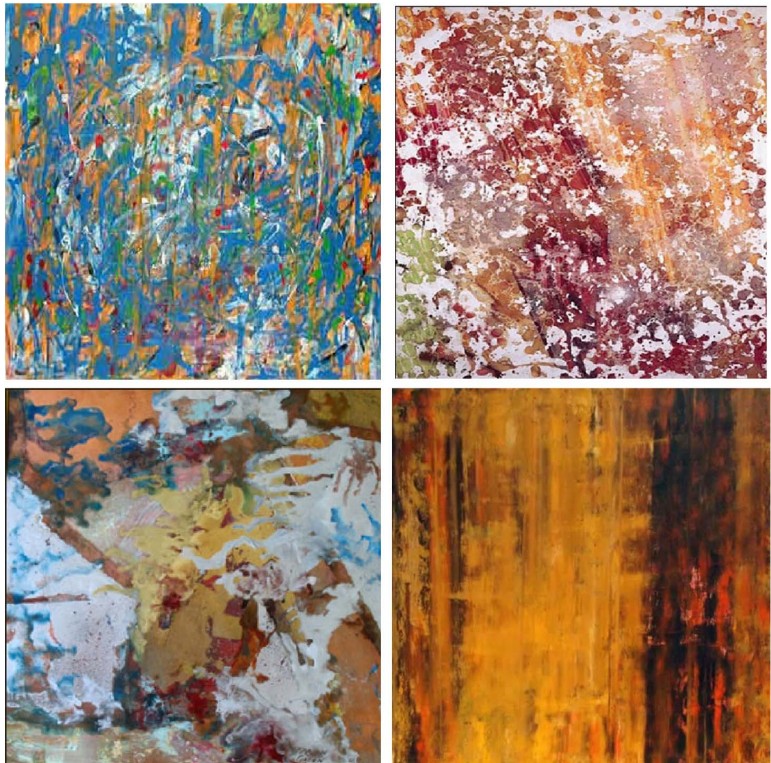

**Fig 1. Examples of stimuli used in both experiments.**

## Procedure

On entering the lab, participants were told they would be presented with a series of paintings, digitized, and presented on a computer display, which would be preceded by a label indicating the general category of the artwork. They were asked to view each work in the context of the label. Consent was then secured, and participants were placed in the eye-tracker with their heads on a chin rest to minimize any head movements during the experiment. After camera set up, the room was darkened to almost complete darkness, barring that from the computer display. Participants' eyes were then calibrated using a nine-point calibration commonly used with the SR Research Eyelink II eye tracking system before proceeding with viewing the artworks. Calibration was only accepted when there was an overall difference between initial calibration and a validation retest of less than 0.5 degrees of visual angle. In the event of a failure to validate, calibration was repeated. If necessary, camera setup was repeated before re-calibration. Eye movement responses were recorded from the left eye only, although the right eye camera was placed as if it was recording. Recalibration took place after every 10 trials. During the experiment, eye movement responses were monitored on a dedicated eye tracker output display computer monitor to ensure that participants engaged with the task and did look at both the label and the artwork.

## Data analysis

Saccade and fixation events were detected using SR Research Data Viewer software. Saccades were defined as periods when the eye was computed to be moving with a velocity greater than 22 degrees per second and an acceleration greater than 8000 degrees per second $^2$. Fixations were defined as occasions when the eye was not moving saccadically. No

minimum duration was set. From this, we extracted average fixation counts by taking the average of the total number of fixations made on each trial for each participant before averaging across participants. Average fixation duration (the converse of saccade latency and saccade duration) was derived in the same way. Saccade amplitude was derived from the eye movement recordings and was defined as the distance between the start and end of the saccade in degrees of visual angle.

The predicted effects of art content labels on these gaze parameters were common across both experiments, but there were some different predicted effects in each experiment. Saccade direction was extracted for Experiment 1 as it was predicted that saccade direction would be affected by the art category labels of Landscape and Portrait, such that each would elicit more horizontally directed or vertically directed saccades, respectively. Saccade direction is the angular component of the saccadic eye movement and, for this experiment, was grouped into broad categories of horizontal and vertical Those saccades whose direction was less than 45 angular degrees from the horizontal meridian were labelled as such, and those outside of this range were labelled vertical. From this saccade, direction counts were computed. For Experiment 2 it was predicted that the art category labels of Still Life and Action would impact the dynamics of the saccadic eye movement such that those made when viewing art labelled as action would be more dynamic: Those labelled Action would elicit shorter duration saccades with higher velocities. To examine this saccade duration and their peak velocities were extracted for Experiment 2.

The experiments were designed under a frequentist approach and consequently, data were analysed using Analysis of Variance (ANOVA) with factors of Label and/or Time as detailed in the Results section. To supplement this a series of Bayesian repeated measures ANOVAs are reported with subject and all repeated measures as random slopes for all dependent measures (JASP version 0.19.1; jasp-stats.org). Bayes factors are a way to measure the relative support that data provide to competing hypotheses. These were computed using the default prior options for the effects within JASP (i.e., r scale = 0.5 for the fixed effects; r scale = 1 for the random effects and r scale = 0.354 for the covariates). Bayes factors ($BF_{10}$) are reported here. These express the probability of the data given H1 relative to H0 (i.e., values larger than 1 are in favour of H1), to indicate evidence for models of the data that include main effects and/or their interaction terms. These were organised to indicate the predictive performance of each model compared with the best model. Based on Jeffreys' [51] criteria [see [52]] $BF_{10}$'s are considered anecdotal (ambiguous) evidence for factor inclusion if between 1 and 3; moderate supporting evidence if between 3 and 10; and strong if between 10 and 30; very strong if between 30 and 100; extreme if greater than 100.

## Results

To help the reader visualize the eye movements we recorded example plots of recorded scan paths. These are shown in Fig 2 overlaid upon the abstract artwork that evoked that scan path response. Each artwork and scan path shown is a single trial from different participants. Those on the left are from Experiment 1 (labelled for each participant as: "Landscape", "Portrait" and "Abstract" from top to bottom) and those on the right are from Experiment 2 (labelled "Still Life", "Action" and "Abstract" from top to bottom). Plots show fixation locations as blue circles along with the start and end points of each saccade (with a joining straight yellow line). Fixation durations are represented by circle diameter with wider ones indicating longer durations as shown by the number next to each circle which shows the actual duration in milliseconds. The location of the saccade in the scan path sequence is given the yellow number towards the end of each line, e.g., 11 would mean the 11th saccade in that scan path sequence for that trial.

### Experiment 1

Table 1 shows the average number of fixations made and their average duration for each label type. The average saccade amplitude is also shown. Repeated measures standard error for each is shown in brackets [53]. Repeated measures ANOVA with label type as the factor shows no effect on these measures (F's < 1.2).

# Experiment 1    Experiment 2

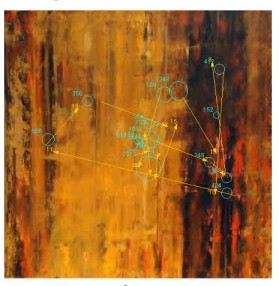
Landscape

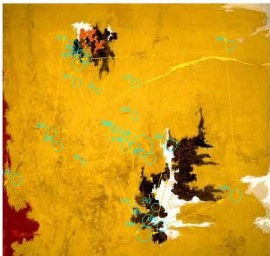
Still Life

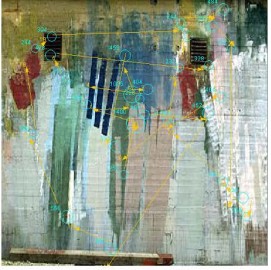
Portrait

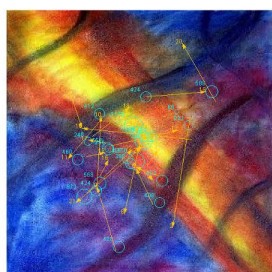
Action

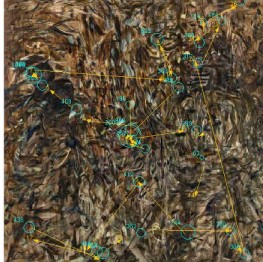
Abstract

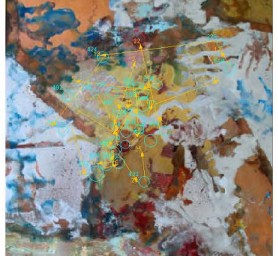
Abstract

**Fig 2. Shows example scan paths for six abstract art works for six different participants.** The three on the left are from Experiment 1 (labelled for each participant as: "Landscape", "Portrait" and "Abstract" from top to bottom) and those on the right are from Experiment 2 (labelled "Still Life", "Action" and "Abstract" from top to bottom). Fixations are plotted as pale blue circles, the center of each is the fixation location while their diameter represents their duration which is also given by the number (in milliseconds) next to each circle. Saccades are shown as straight yellow lines between their start and end points with the yellow number indicating the point at which each saccade as executed in the path sequence (i.e., 1 is the first saccade, 2 is the second and so on).

**Table 1. Overall average saccade and fixation parameters are shown for each art label type for Experiment 1 and 2.**

| | Experiment 1 | | | Experiment 2 | | |
|---|---|---|---|---|---|---|
| | **Abstract** | **Landscape** | **Portrait** | **Abstract** | **Still** | **Action** |
| **Fixation Count** | 12.8 (.36) | 12.5 (.38) | 12.6 (0.32) | 20.8 (.6) | 20.7 (1.05) | 20.6 (1.19) |
| **Fixation Duration** | 368.3 (12.5) | 380.2 (14) | 372.6 (11.4) | 375.7 (13.9) | 380.9 (19.7) | 379.3 (22.6) |
| **Saccade Amplitude** | 3.42 (.12) | 3.39 (.12) | 3.36 (.12) | 2.0 (.1) | 2.0 (.1) | 2.0 (.1) |
| **Saccade Duration** | | | | 39.9 (3.5) | 41.9 (3.2) | 45.8 (5.5) |
| **Peak Velocity** | | | | 168.9 (16.7) | 158.8 (12.1) | 167.0 (20.7) |

Standard errors are shown in brackets. Fixation count, duration and saccade amplitude are shown for both experiments. Saccade duration and peak velocity were additionally extracted for Experiment 2 to examine specific predictions about the effect of Action as a label on saccade dynamics. Fixation and Saccade Duration are shown in milliseconds. Saccade Amplitude is shown as degrees of visual angle. Saccade Peak Velocity is shown as degrees per second. Error cars show repeated measures estimates of error [53].

To examine average viewing behaviour throughout the trial, fixation and saccade measures were extracted for the first 2 seconds of viewing, the middle period from 2 to 7 seconds and the final period from 7 to 15 seconds. This matches those periods identified as the gist phase and survey phase (Locher, 1996; Locher et al, 2007; Nodine & Krupinski, 2003; Velichkovsky et al., 2005). The outcome of this process is shown in Fig 3.

To derive the average saccade and fixation parameters generally, they were first averaged for each trial and then further averaged for each label for each participant. For average fixation counts, fixations were totalled for each trial and

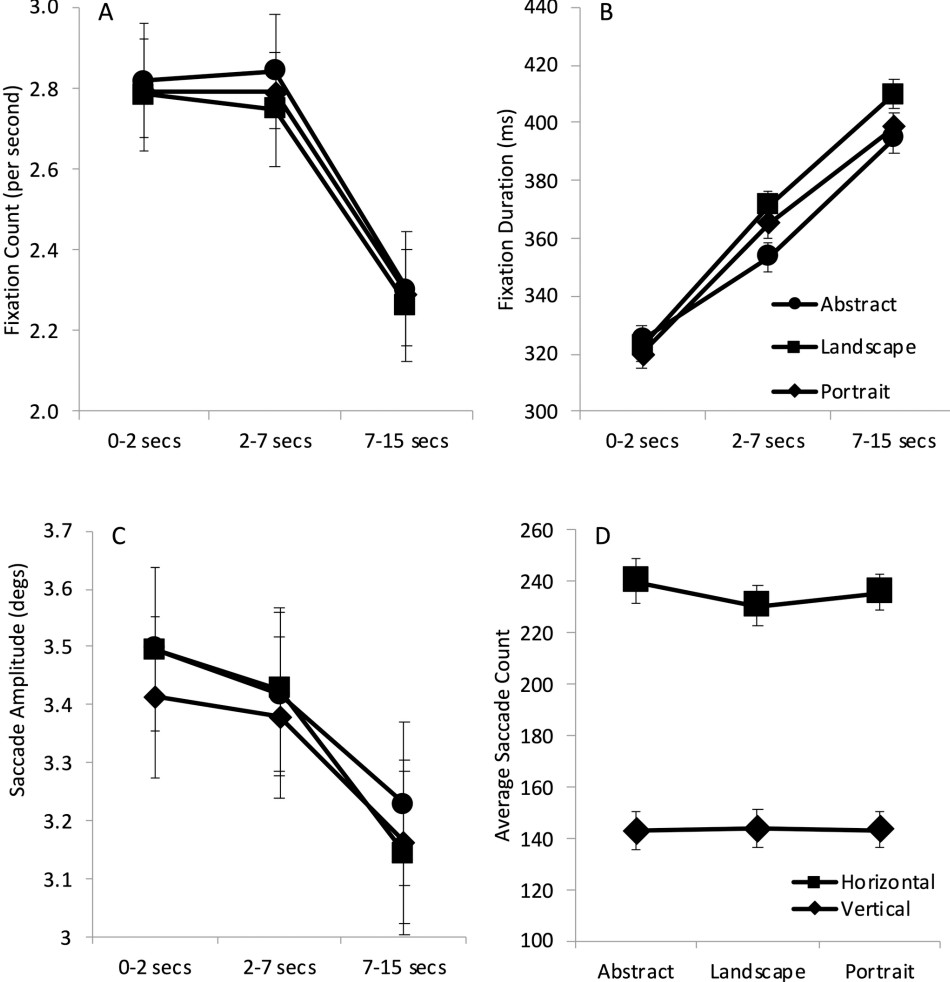

**Fig 3. Shows eye movement parameters across the viewing period.** The upper row from left to right shows **A** average fixation counts and **B** average fixation durations. The lower row, left, shows **C** average saccade amplitude in degrees of visual angle. These show little effect of art content label but clear effects of viewing time. The Lower row, right, shows the **D** average saccade counts for those classed as "horizontal" and "vertical" as a function of the label. Again, little effect of art content label is seen, but there is a clear bias for horizontally directed saccades. Error cars show repeated measures estimates of error [53].

then averaged across trials. To compensate for the fact that more fixations were made in wider time bins, the number of fixations made (fixation count) was divided by the number of seconds in each bin. This compensation was not applied to any other parameter.

Fig 3A shows the average fixation count across participants. A two-way repeated measures ANOVA with Category Label (abstract, landscape, portrait) and Time (0–2 secs, 3–7 secs, and 7–15secs) as the two factors shows a significant main effect of Time only (Time: $F(2, 58)=64.319$, $p < .001$, $\eta^2 = .689$; other $p$'s $> .364$). This is also apparent from the Bayes repeated measures ANOVA analysis, which shows that the best predictive model of the data included only Time as a factor with the next best model, of both main effects of Time and Label, being 0.165 ($BF_{10}$) much less likely. The null model was extremely unlikely compared with Time alone ($BF_{10}=7.789 \times 10^{-13}$) and the addition of Label only was 0.16 moderately in favor of the null model (i.e., Label $BF_{10}$/Null model $BF_{10}=1.241 \times 10^{-13}/7.789 \times 10^{-13}$). Contrasts show that fixation counts in the final period are significantly lower than the earlier period and Bayesian post hoc analysis favors an effect of Time on fixation counts (0–2 secs vs 2–7 secs, $t(29)<1$, $BF_{10}=0.118$; 2–7 secs vs 7–15 secs, $t(29)=11.113$, $p < .001$; $d = 1.08$, $BF_{10}=7.541 \times 10^{+22}$).

In line with the decrease in fixation counts, fixation durations (Fig 3B) show a steady increase as viewing progresses regardless of Category Label (Time: $F(2, 58)=33.450$, $p < .001$, $\eta^2 = .661$, all other $p$'s $> .600$). Bayes analysis also shows Time as the best model and the addition of Label being 0.164 times less likely. The null model was extremely unlikely compared with Time alone ($BF_{10}=1.827 \times 10^{-8}$) and, again, the addition of Label was 0.165 in favor of the null model (i.e., Label $BF_{10}$/Null model $BF_{10}=3.022 \times 10^{-9}/1.827 \times 10^{-8}$). Contrasts show that this was a significant increase across time bin in comparison to the proceeding one with Bayes factors providing extremely strong evidence for an effect of Time (0−2 secs vs 2−7 secs, $t(29)=-6.614$, $p < .001$, $d = -.558$, $BF_{10}=6.094 \times 10^8$; 2−7 secs vs 7−15 secs, $t(29)=-4.004$, $p < .001$, $d = -.511$, $BF_{10}=26983.349$).

Saccade amplitude was found to decrease over time with a significant main effect of Time only (Time: $F(2, 58)=8.341$, $p < .001$, $\eta^2 = .223$; other $p$'s $> .475$; Fig 3C). As with fixation count and duration, Bayes ANOVA analysis shows Time as the best model, with the addition of Label being 0.103 less likely. The null model was very unlikely compared with Time alone ($BF_{10}=0.02$), and the addition of Label was moderately in favor of the null model (i.e., Label $BF_{10}$/Null model $BF_{10}=0.002/0.02=0.1$). Contrasts show saccade amplitude in the final period was significantly longer than the earliest period, and Bayes post hoc tests also provide extremely strong evidence for shorter saccade amplitudes in the final period (0–2 secs vs 2–7 secs, $t<1$, $BF_{10}=0.19$; 2–7 secs vs 7–15 secs, $t(29)=4.592$, $p < .001$; $d = .323$, $BF_{10}=21765.38$).

We hypothesized that labels of landscape or portrait would affect saccade direction with landscape images evoking more horizontally oriented movements, while those works described as a portrait might provoke more vertical movements both relative to each other and to the abstract labelled works. Saccades were broadly defined as horizontal and vertical (See Methods), and their frequency of occurrence was examined using a two-way repeated measures ANOVA with Label type and Orientation as factors (see Fig 3D). This showed a significant effect of Orientation only with participants making more horizontal movements recorded regardless of the artwork description ($F(1, 27)=85.681$, $p < .001$, $\eta^2 = .747$; other $p$'s $> .271$). Bayes analysis shows Orientation as the best model and the addition of Label being 0.153 less likely. The null model was extremely unlikely compared with Orientation alone ($BF_{10}=5.315 \times 10^{-7}$), and the addition of Label was 0.146 in favor of the null model (i.e., Label $BF_{10}$/Null model $BF_{10}=7.769 \times 10^{-9}/5.315 \times 10^{-7}$).

Overall, no effect of the categorical superordinate label of Landscape or Portrait was found on viewing patterns relative to that found with those labelled Abstract. Indeed, when taken across all eye movement measures reported here, Bayesian analyses show convincing evidence against there being any effect of Category Label on viewing behavior.

## Experiment 2

Overall eye movement and fixation parameters are shown in Table 1 for each art label type. As with Experiment 1, very little effect of Category Label was found for any of these (all $F$'s $< 1$). Fig 4 shows average viewing behaviour throughout

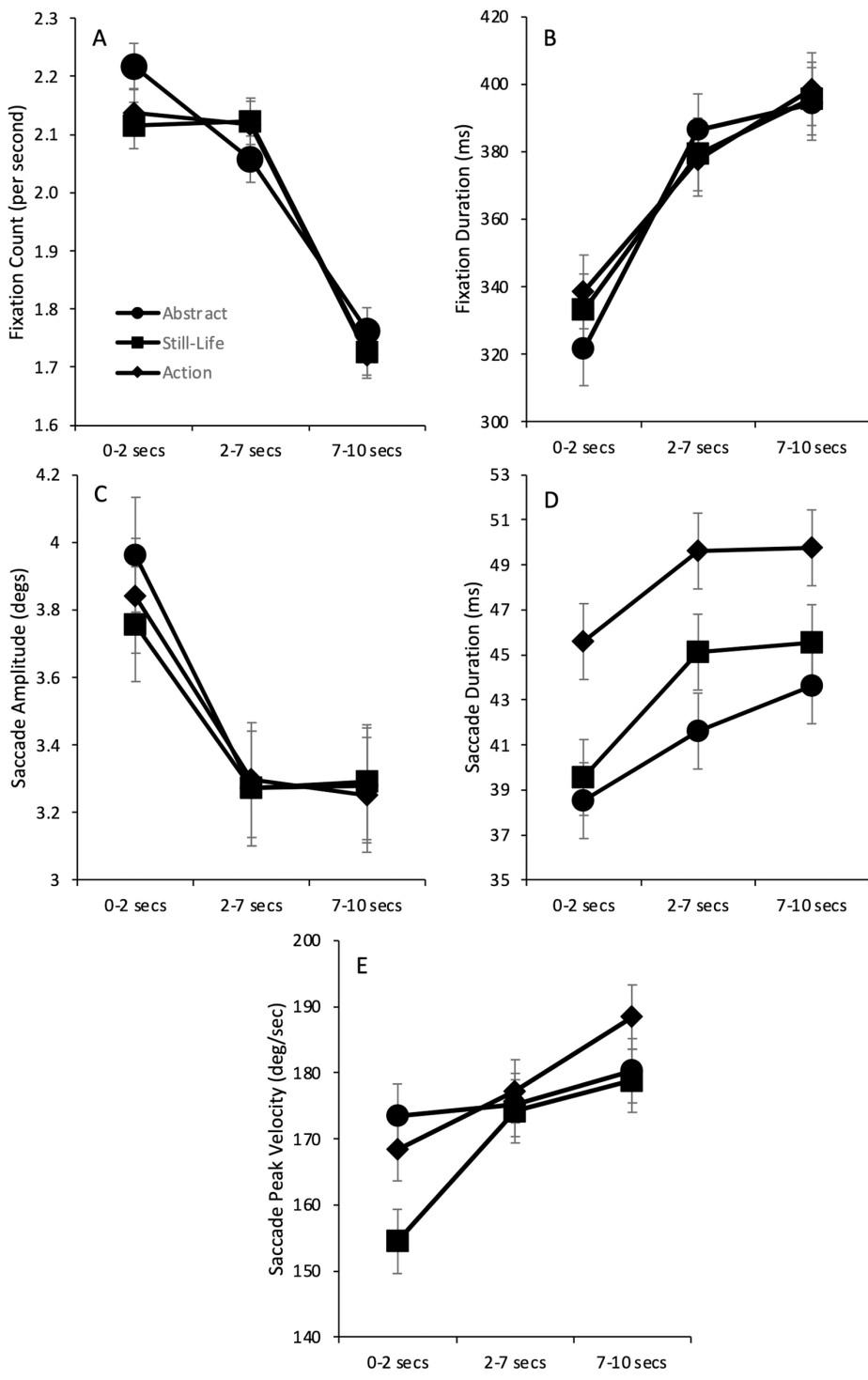

**Fig 4. Shows eye movement parameters across the viewing period for each categorical label (Abstract, Still life, and Action).** The upper row from left to right shows **A** average fixation counts and **B** average fixation durations. The middle row shows **C** average saccade amplitude in degrees of visual angle and **D** shows the average saccade duration in milliseconds (ms). The lower row **E** shows **the** average saccade peak velocity in degrees per second. Each of these eye movement parameters shows little effect of CategoryLabel but clear effects of viewing time. Error cars show repeated measures estimates of error [53].

the trial for fixation and saccade parameters shown in Table 1. Measures were extracted for the first 2 seconds of viewing, the middle period from 2 to 7 seconds and the final period from 7 to 10 seconds.

As with Experiment 1, fixation counts were totalled for each trial and the average was drawn for each label separately for each participant. To compensate for the fact that more fixations were made in wider time bins, the number of fixations was divided by the number of seconds in each bin. Fig 4A shows the average fixation count across participants. A two-way ANOVA with Category Label (Abstract, Still life and Action) and Time (0–2 secs, 3–7 secs, and 7–10 secs) as the two factors shows a significant main effect of Time only ($F(2, 42)=121.4$, $p<.001$, $\eta^2=.853$; other p's>.149). This is also apparent from the Bayes repeated measures ANOVA analysis, which shows that the best model of the data included only Time as a factor with the next best model, of both main effects of Time and Label, being 0.508 ($BF_{10}$) less likely. The null model was extremely unlikely compared with Time alone ($BF_{10}=9.2\times10^{-15}$), and the addition of Label was moderately in favor of the null model (i.e., Label $BF_{10}$/Null model $BF_{10}=2.004\times10^{-15}/9.2\times10^{-15}=0.22$). Contrasts show that counts in the final period are significantly lower than in the earlier period, and Bayes post hoc factors favor the effect of Time on fixation counts (0–2 secs vs 2–7 secs, $t(21)=1.901$, $p=.071$, $d=.131$, $BF_{10}=0.754$; 2–7 secs vs 7–15 secs, $t(21)=15.083$, $p<.001$; $d=.827$, $BF_{10}=7.945\times10^{19}$).

In line with there being fewer fixations, fixation durations show a steady increase as viewing progresses regardless of Label (Fig 4B; $F(2, 42)=40.997$, $p<.001$, $\eta^2=.661$, all other p's>.600). This is also apparent from the Bayes repeated measures ANOVA analysis which shows that best model of the data included only Time as a factor with the next best model, of both main effects of Time and Label, being 0.247 ($BF_{10}$) less likely. The null model was extremely unlikely compared with Time alone ($BF10=3.798\times10^{-8}$), and the addition of Label was 0.25 moderately in favor of the null model (i.e., Label $BF_{10}$/Null model $BF_{10}=9.655\times10^{-9}/3.798\times10^{-8}$). Contrasts show that fixation durations are significantly longer than earlier periods, with Bayes factors also favoring this interpretation (0−2 secs vs 2−7 secs, $t(21)=-6.117$, $p<.001$, $d=-.650$, $BF_{10}=1.977\times10^7$; 2−7 secs vs 7−15 secs, $t(21)=-2.292$, $p<.032$; $d=-.195$, $BF_{10}=3.351$).

Saccade parameters (Fig 4C-4E) also show a main effect of Time only. To take each in turn. Saccade amplitude showed a significant main effect of Time ($F(2, 42)=77.064$, $p<.001$, $\eta^2=.784$; other p's>.408) with specific contrasts showing that saccade amplitude in the later periods was significantly shorter than the earlier period, with no differences found between the mid and later periods (0−2 secs vs 2−7 secs, $t(21)=9.835$, $p<.001$, $d=1.041$, $BF_{10}=1.717\times10^{12}$; 2−7 secs vs 7−10 secs, $t<1$, $BF_{10}=0.137$). Both saccade duration and peak velocity also showed significant main effects of Time (Saccade Duration: $F(2, 42)=3.219$, $p=.050$, $\eta^2=.133$; other p's>.394; Peak Velocity: $F(2, 42)=4.071$, $p=.024$, $\eta^2=.162$; other p's>.294). Further contrasts show that there was some anecdotal evidence that both saccade duration and peak velocity were greater in the later periods than in the earlier periods (Saccade Duration: 0−2 secs vs 2−7 secs, $t(21)=-.1884$, $p=.037$ (1-tailed), $d=.194$, $BF_{10}=1.519$; 2−7 secs vs 7−10 secs, $t<1$, $BF_{10}=0.175$; Peak Velocity: 0−2 secs vs 2−7 secs, $t(21)=2.099$, $p=.048$, $d=-.116$, $BF_{10}=2.366$; 2−7 secs vs 7−10 secs, $t(21)=-1.649$, $p=.114$, $d=-.08$, $BF_{10}=0.669$).

Bayesian Repeated Measures ANOVAs across all saccade parameters show Time as the best model of the data with no support for the next best model, either the additional main effect of Label (Saccade amplitude: $BF_{10}=0.199$; Peak velocity: $BF_{10}=0.514$), or the null model for saccade duration ($BF_{10}=0.92$; Main effect of Label: $BF_{10}=0.451$). The addition of Label was also anecdotal to moderately in favor of the null model compared with the model including the main effect of Label (i.e., Label $BF_{10}$/Null model$_{10}$; Saccade Amplitude: $6.122\times10^{-12}/2.789\times10^{-11}=0.22$; Saccade Duration: $0.451/0.92=0.49$; Peak Velocity: $0.216/0.482=0.448$).

Overall, in Experiments 1 and 2, fixations were more concentrated in later viewing periods with fewer being engaged and their durations increased. Saccades between fixations show a decrease in amplitude, with saccade durations and their peak velocity all found to increase over the viewing period as fixations decrease. This pattern mirrors that commonly reported. Viewers' scan paths, when faced with visual environments from scenes to artworks shift from a pattern of eliciting many large amplitude saccades with short fixations executed to a more concentrated series of eye fixations. Our interpretation, along with many others, is that this serves to support the extraction of an initial holistic gist of the artworks

prior to a particular subsequent survey of areas of interest for further scrutiny. The pattern is very obviously supported by consideration of the decrease in fixation counts, the increase in their durations, and the reduction of saccade amplitude magnitude as participants spend more time viewing the artworks. There was found to be no effect of categorical labels on this viewing pattern for any of the saccade or fixation parameters, suggesting that high-level superordinate categorical labels describing the general content of the works did not affect viewers' explorations of the pieces. The Bayes analysis presented provides convincing evidence for this interpretation.

## General discussion

We hypothesized that eye movement patterns would show a progression from gist-ambient viewing to survey-focal style viewing with early eye movements showing large amplitude saccades separated by short periods of fixation and those made later showing a pattern of small amplitude saccades and longer fixations. We further hypothesized that the broadly specified categorical labels given to the artworks would affect this gaze pattern. In line with the first hypothesis, gaze was found to change over the viewing period in the manner predicted, however, the categorical labels were quite convincingly shown to not affect this pattern. Across all categorical labels fixation counts were generally found to diminish, and their durations lengthened in the later periods of viewing the artworks. Fixation locations were also closer together in this latter part of viewing, as shown by the decrease in saccade amplitude over time. This change in eye movements and gaze preponderance throughout viewing shows a pattern of viewing which supports an initial broadly spread and rapid scan of abstract artworks which then settles down into greater scrutiny of select areas of the works. This matches the Gist-Ambient and Survey-Focal phases of art viewing [15,19,20]. This shift from the initial extraction of holistic information about the works followed by more detailed scrutiny marries well with the temporal progression through a series of stages outlined by current models of art perception [23,24,54]. In these models, the viewer starts with a pre-classification of the artworks in which their understanding and knowledge of it informs their "artistic design stance" and subsequently their "artistic understanding". Expectations [55,56], personal relevance [57] and prior affective state also play a role at this point [24,58]. On viewing the works, automatic initial early visual processing serves to extract information about the gist of the works, the style, orderliness, complexity, grouping, and texture guiding low-level attention and eye movements to areas of interest. Higher-level visual representations would then integrate with prior knowledge held in memory, for instance, its familiarity, or its prototypicality. This then allows an explicit classification of style, content, and artistic techniques used. And finally, cognitive mastering in which elements extracted from the works are combined to create meaning, to produce an aesthetic response and an aesthetic judgement. The time course of these stage-by-stage models is such that each stage feeds into the next in a feedforward manner, with feedback also influencing those stages that come beforehand. Furthermore, these models allow for stages to be revisited and cycled through again as needed to broaden and deepen aesthetic response and understanding. For example, a model put forward by Leder and colleagues [24] and recently reviewed alongside other models by Pelowski et al (*see* [25,59] has each stage cycled through quite quickly suggesting that initial aesthetic response and judgement can be reached in a range from 1s to 6s.

In Experiment 1, we also suggested that more horizontally oriented movements would be elicited for works labelled as Landscape, and more vertically oriented for those labelled as Portraits both relative to each other and to those labelled as Abstract. Neither of these patterns was found. Participants generally made more horizontally oriented movements than vertically oriented ones regardless of categorical labels. This natural difference in the distribution of saccade direction is in line with a previously reported bias for horizontal eye movements across a variety of tasks and stimuli [60–63] and can be explained by oculomotor, perceptual and external factors. Oculomotor factors include the dominance of muscular or neural mechanisms which preferentially produce horizontal movements of the eyes regardless of the low-level visual features of the stimuli. Furthermore, vertical and oblique saccades have been reported to be slower and moved curved in trajectory [64–66], And horizontal saccades of greater than 5 degrees have shorter response latencies, higher peak velocities and better accuracy than vertically directed ones [67–70]. The second factor which may result in more horizontally oriented

eye movements may be related to the finding that visual perception of stimuli is better along the horizontal meridian than the vertical. This has been reported for orientation and contrast discrimination [71,72], spatial frequency discrimination [73], visual crowding [69,74], letter identification [75] and spatial localization [69], and has been linked to anisotropies in spatial coding in spatial short-term memory [76]. The third factor at play may be the distribution of features in natural scenes. The presence of the horizon gives a strong horizontal contrast edge and there tends to be a clustering of salient features near the horizon [77]. Stimuli commonly used in many tasks show a non-uniform distribution of salient features due to composition factors, e.g., objects of interest are commonly placed centrally in photographs [78] which could produce more horizontal movements through learned top-down associations. Given this, it may be the case that when faced with abstract artworks viewers elect to default to a common pattern of scanning the artworks with a greater preponderance of horizontal saccades and fixations compared with vertically directed ones. This would serve to maximise information pick-up along the axis for which we have the greatest sensitivity, are likely to produce faster saccades with better accuracy and are more likely to observe useful clusters of interesting and important features.

For Experiment 2, we hypothesized that the categorical labels of Action and Still Life would be reflected in eye movement responses when viewing the artworks such that the label Action would elicit more saccades with shorter fixations and higher velocities than those labelled Abstract or Still Life works. We also hypothesized that Still Life labelled works would elicit fewer saccades with longer periods of fixation and lower saccade velocities relative to either the Action or Abstract categories. We did not find any support for these predicted differences in eye movement control. the evidence was either ambiguous or argued against an effect of these categorical labels on eye movement control. Previous work has reported that artworks rated as being more dynamic are viewed with a greater number of saccades, with higher velocities and fewer fixations with shorter durations than those rated less highly [79,80]. This was suggested to be the result of the direct impact of the perceived dynamism of the works on viewing behaviour. However, no such difference was found here when inducing different viewing expectations or predispositions in the viewer using a categorical label and abstract artworks. Rather, it is the abstract artwork itself and its inherent visual properties that dictate the viewing behaviour found here. This is perhaps understandable in terms of the suggested effects on viewers when viewing abstract art. The effect of representational art and abstract art on the viewer has been the subject of art theory since the 18th century [81]. For representational work, the effect on viewer response and behaviour can be directly linked to the subject(s) of the artwork and the elements used to create it. Within the context of abstract art, however, it is assumed that its effect on viewers is caused by the use and manipulation of the elements of works: its use of lines, colors, and forms [82]. For example, some works of abstract art are described as more calming and static, such as those by Piet Mondrian or Josef Albers, while others are described as chaotic or dynamic, such as those by Jackson Pollock or Piero Manzoni [82]. These aspects of the works have been suggested to directly influence the way viewers look at the paintings. For instance, some artworks described as dynamic are considered to be so because of the intention and dynamic actions of the artists. Indeed, some have gone so far as to suggest that action paintings provoke action viewing and that without action viewing, there is no action painting [83]. This has been linked to the suggestion that the viewer's response to artworks is an embodied one in which their bodily responses reflect the actions of the artists, possibly via a network of mirror neurons [84,85]. A further suggestion has been made that dynamic paintings elicit a broader distribution of attention due to the lack of a small number of specific areas of interest in the works [86,87]. The works are decentralized, polyphonic and polyfocal. Thus, eye movements would be expected to be spread more widely than those evoked by static paintings. Indeed, it has been suggested that for dynamic action paintings, eye movements would be more common as there is nowhere to settle for extended periods and it is this that elicits the perception that the works are vital and dynamic [88,89]. The term "polyfocal all-over" has been suggested to describe this pattern of behaviour on elements in works, such as those by Pollock, that are homogenous and have no obvious centre within them [90]. Support for this has been reported by [91]. They found that the distribution of fixations was much broader for more dynamic paintings compared with those rated as less dynamic although fixation duration was found not to differ. Whereas [79], using abstract works associated with descriptions of static

and dynamic, found that those rated more highly for dynamism evoked an increased number of fixations with shorter durations, a greater number of saccades, with faster saccade velocity and increased ratings of pleasantness. Given this, it follows that no effect of categorical label was found here because it is the painting itself which evokes the eye movement response with more dynamism perceived in the works eliciting more dynamism in viewing behaviour. The top-down expectation of content has little or no role to play in evoking more or less "action viewing" behaviour.

A further consideration in the control of gaze when viewing artworks is a potential role of stable individual differences in viewing behavior and the shift from the Gist-Ambient to the Survey-Focal phase of art viewing. Measures of oculomotor behavior exhibit stable characteristics that can be captured with common descriptive statistics, but they also reveal a wide range in individual responses. This can be seen in large population studies of saccades, anti-saccades and smooth pursuit [92–95]. Despite these dissimilarities, recent research has identified stable gaze patterns within individuals. Across a range of diverse tasks individuals produce similar eye movement behaviours such as fixation numbers, fixation durations, and saccade amplitudes [96–101], with a similar consistency also shown in measures of micro-saccadic eye movement behavior such as fixation extent, micro-saccade number and amplitude [102]. Recently Balgary et al [103] reported reliable individual pro- and anti- saccade and smooth pursuit eye movement patterns across task and test-retest sessions with eye movement parameters that correlated well within eye movement type. These have been likened to an oculomotor signature [103] and have been related to differences in personality traits [104] and genetic influences [105]. Poynter et al [102] identified strong correlations across six eye movement metrics (e.g., fixation rate, duration and size, saccade amplitude, microsaccade rate and amplitude) that were captured by a single underlying latent factor. Zangrossi et al [106] also reported that viewing behavior dynamics of a large sample of participants in response to a wide range of real-world natural scenes was explained well by very few latent variables with only three components explaining about 60% of eye movement dynamics (those that more reflect how or when people look: fixation duration, and number, gaze step direction and amplitude of saccade). Unlike gaze dynamics analysis of the spatial distribution of fixations (where people look: density map, semantic map and saliency map) show a dependency of gaze distribution on image meaning and points of saliency. The low dimensionality of gaze dynamics led Zangrossi et al [106] to identify individual viewing styles into the two stable types of static and dynamic viewers, that reflective automatic intrinsic, endogenous preferences that are executed relatively independently of differences in the external visual environment and remain stable when participants view a blank screen. These viewing styles have been linked to persistent differences in resting-state EEG brain activity [107] shown in static and dynamic viewers tested one year after behavioral experiments reported by Zangrossi et al [106]. These differences in resting-state have been linked to higher cortical inhibition and focus on internal processing for static viewers while dynamic viewers have a profile more biased toward cortical excitation and external processing. This interpretation is bolstered by behavioral outcome measures with static viewers showing slightly stronger visual working memory and dynamic viewers showing weaker inhibition of salient but non-relevant stimuli. These stable individual differences in eye movement control and viewing behaviours are likely to also play a role in gaze patterns while viewing artworks and in the eye movement outcomes reported here. This is especially likely when the stimuli viewed were abstract artworks which may encourage eye movements that reflect an individual's idiosyncratic endogenous internally driven viewing style rather than being influenced by the artwork's semantic content, something that would be found in more structured paintings such as portraits or landscapes. This is something that would be fruitful to more thoroughly explore in future studies in this area.

## Practical implications

We are hesitant to offer practical implications for artists and those who seek to improve accessibility to art. However, it is notable that art viewers like to have labels and context to artworks which is most obviously shown by the cyclical pattern of gaze: first art – then label – then back to art gaze behaviour recorded in museum visits [8,9]. If label information is available most viewers will use it. The research outlined in this article, and the outcomes of the study reported here, suggest

that descriptive titles [28–31] and broader categorical information will not affect gaze behaviour as the gist of an artwork is readily extracted in very few fixations. However, elaborative titles and information about an artwork which serve to reduce ambiguity [29] or highlight its more thematic and interpretative aspects [26] will be used and will impact gaze patterns and verbal descriptions. This suggests that if you are an artist or curator (for example) and you want your audience to be drawn to the aspects of the work not captured by the visual structure and semantic meaning then tell them about it.

## Conclusion

In conclusion, the results show that saccadic eye movements made in response to abstract artworks are initially widely spread, and fixations are brief, but over time, saccades become less frequent, and fixations are more focussed on areas of interest. This pattern of saccades and their fixation periods is not affected by contextual information given by preceding categorial labels given to the works, suggesting a minimal role for top-down control of eye movements afforded by these labels when viewers are faced with abstract artworks. It has been suggested that titles only contribute to understanding and liking if they form part and contribute to a rich, successful, and coherent representation [108] which may be part of a wider effect of contextual information that acts to promote greater fluency [109]. This facilitates the processing of the artwork, reduces mental effort, and consequently leads to a greater understanding and perhaps greater liking of the work. We interpret the changing pattern of gaze behaviour when viewing artworks reported here as reflecting a temporal evolution from a Gist-Ambient mode of viewing to a more Focal-Survey mode of viewing. Initial eye movement responses support automatic bottom-up, low-level processing of the visual features of artworks to establish their gist. This is succeeded by the more focussed gaze pattern survey of the important areas of the artworks in more detail to support meaning-making and scaffold a deeper understanding the works.

## Supporting information

**S1 File. All in-house paintings created and used as stimuli are shown here.**
(DOCX)

## Acknowledgments

Many thanks to Lily House, Anya Richards and Natalie Rosset for help with preparing the materials and collecting the data for the study.

## Author contributions

**Conceptualization:** Eugene McSorley.

**Data curation:** Eugene McSorley.

**Formal analysis:** Eugene McSorley.

**Investigation:** Eugene McSorley.

**Methodology:** Eugene McSorley.

**Project administration:** Eugene McSorley.

**Resources:** Eugene McSorley.

**Software:** Eugene McSorley.

**Supervision:** Eugene McSorley.

**Writing – original draft:** Eugene McSorley.

**Writing – review & editing:** Eugene McSorley, Rachel McCloy, Louis Williams.

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
