## [Decision Letter · Decision Letter 0]

Dear Dr. McSorley,

Thank you for submitting your manuscript to PLOS ONE. After careful consideration, we feel that it has merit but does not fully meet PLOS ONE’s publication criteria as it currently stands. Therefore, we invite you to submit a revised version of the manuscript that addresses the points raised during the review process.

We look forward to receiving your revised manuscript.

Kind regards,

Hosam Al-Samarraie

Academic Editor

PLOS ONE

Journal Requirements:

Additional Editor Comments :

Dear authors, the work is relevant and well justified. I would advise you to review the comments below and respond to them for reconsideration. In addition to the reviewer's comments, please see mine below:

The argument in the introduction is well established and linked to the goals of Exp1 and 2. However, you might want to include some recent examples from previous studies on viewing behaviors of different styles of painting and their results. This can be added before “We suggest that the reason why descriptive titles….”

You did mention some studies (Early eye movements reflect the structural elements and semantics of art works, its gist, with…) but not specifically explaining them for readers to be able to see the difference to yours.

In the method, have you thought of using mixed model analysis? This type is more effective in handling smaller sample sizes and controlling for type I errors by considering random effects and fixed effects together. Can you please explore the possibility of integrating this to your current study.

Reading the discussion, it is a bit difficult to extract the practical implications of this work. I would suggest that you add this as a separate section after the discussion.

The limitation and conclusion in the discussion can be separated into a separate section.

Reviewers' comments:

Reviewer's Responses to Questions

**Comments to the Author**

1. Is the manuscript technically sound, and do the data support the conclusions?

Reviewer #1: Yes

2. Has the statistical analysis been performed appropriately and rigorously?

Reviewer #1: N/A

3. Have the authors made all data underlying the findings in their manuscript fully available?

Reviewer #1: Yes

4. Is the manuscript presented in an intelligible fashion and written in standard English?

Reviewer #1: Yes

Reviewer #1: This is the review for manuscript PONE-D-24-31271 entitled: “Viewing of abstract art follows a gist to survey gaze pattern over time regardless of broad categorical titles”. Therein, the authors describe two experiments in which they showed pieces of art to participants while they recorded their eye movements. Prior to the presentation, art works were labelled. The authors assumed that labelling will affect gaze behavior during viewing the artwork. However, this was not the case. The authors argue that the results can be explained by the labels not adding to the processing of the artwork above what would have been expected by a gist to survey viewing behavior and the inherent properties of abstract pieces of art.

Overall, the manuscript is well written, and the analyses are in principal appropriate for answering the research question. The literature review is comprehensive and the explanation of the results plausible. Eye movement recordings and analyses follow the state-of-the-art and are well described. I have some questions and suggestions concerning the analyses. Please, see my major and minor points below.

1) One alternative explanation could be that the labels were ignored by participants. Did the authors check that labels were recognized and processed? For instance, by asking them after viewing about the label of the artwork? If the authors did not check this, I suggest to at least mention this as an alternative explanation for the missing effect of labelling. A stronger manipulation could be to instruct participants if they find the label a piece of art was given suitable.

2) For the frequentist analysis approach, did the authors maker power calculations as to how many participants would be needed to at least find a small or medium effect of the labels? If not, they could calculate the achieved power post hoc and mention it in the Participants section of the manuscript.

3) In frequentist analyses, a null result cannot be interpreted. It would be of interest, if there is convincing evidence against the effect of category label by running a Bayesian ANOVA, for instance with the open source and easy to use software JASP.

4) One suggestion for further analyses. Did the authors compare the same artwork that got different labels for different participants? How did people look on average on one and the same artwork when it received the label landscape or portrait?

Minor

1) While the overall story can be followed, I recommend language proof correction through a native speaker or careful revision of the writing. There are many minor grammar mistakes.

2) Please, also check the accordance to APA style for titles and tables.

**Do you want your identity to be public for this peer review?** For information about this choice, including consent withdrawal, please see our Privacy Policy

Reviewer #1: No

---

## [Author Response · Author response to Decision Letter 1]

19 Nov 2024

PONE-D-24-31271

Viewing of abstract art follows a gist to survey gaze pattern over time regardless of broad categorical titles

PLOS ONE

Dear Dr Hosam Al-Samarraie,

Thank you for your comments and the reviewers comments on our manuscript. Please see below for you detailed responses to the suggestions made.

Best wishes

Dr Eugene McSorley

Journal Requirements:

Response: Checked manuscript throughout and changes have been made accordingly.

Additional Editor Comments :

1. The argument in the introduction is well established and linked to the goals of Exp1 and 2. However, you might want to include some recent examples from previous studies on viewing behaviors of different styles of painting and their results. This can be added before “We suggest that the reason why descriptive titles….”

2. You did mention some studies (Early eye movements reflect the structural elements and semantics of art works, its gist, with…) but not specifically explaining them for readers to be able to see the difference to yours.

In Response to both 1. and 2. above: In response to both of these points we have added further descriptive details of the some of the cited work in the text. We hope this helps illustrate the points made and clarifies the difference between previous work and the study reported here.

Text added to introduction:

“Models such as these make the direct predictions that providing contextual information about artworks will improve meaning making and understanding of the works and will impact on the viewers aesthetic appreciation and experience. To support this, viewer’s gaze patterns when looking at art are affected by the presence of contextual information in the form of titles and descriptions. Fixation distribution across representational artworks has been shown to be wider when contextual information about the artworks is given, and time spent on areas of interest included in the contextual information is increased (26–29). Furthermore, the distance travelled between fixations (saccade amplitudes) is shorter (28) and returns to areas of interest occur more frequently (27). This has been found to differ depending on the style of painting and task undertaken (28–31). For example, when presented with works by Kandinsky, viewers gaze was found to concentrate on those areas indicated in the title (e.g., Painting with white border) and to look at them more often (27). This pattern of viewing was not found for those who were not made aware of the title. Likewise, changing the depth of information from simple, factual, information (including the title, artist name and dates, date of painting and medium and technique) to elaborative descriptions (including the theological and symbolism) of Zurbarán’s paintings of Jacob and this Twelve Sons, showed significant differences in viewing patterns (26). Elaborative information evoked wider fixation distributions and more time spent on areas of symbolic importance. Consistent effects of titles on gaze behaviour are only reported during later viewing periods. For example, viewing behaviour in response to Kandinsky’s painting (27) and Zurbarán’s paintings (26) showed no effect of title on early viewing periods. Title effects only revealed themselves later after the first 2 seconds. Similarly (28,30) reported that, while viewing works by Dali (surrealist such as Swans reflecting elephants) and Caravaggio (baroque such as The sacrifice of Isaac) showed effects of title and task on viewing behaviour, the earliest effects varied depending on the artist style and whether the viewer was asked to give a preference or subsequently describe the work. This fits well with early verbal descriptions of artworks by artists such as Bruegel, Vermeer, and Klee (15), which, along with early eye movements, reflect the structural elements and semantics of art works, its meaning or gist, with effects of contextual information only occurring after the first 2 to 5 seconds of viewing (27). After 7 seconds verbal reactions begin to reflect higher level concepts and comments on art styles, forms and emotional reactions to the works (32).

The lack of consistent effect of titles on early viewing behaviour and early gaze responses and verbal responses to art works suggest that the reason why descriptive titles have been found not to influence early eye movement responses is because they simply describe those aspects of the artworks that are captured by their gist (i.e., the structure and semantic meaning). Being made aware of them prior to viewing an artwork has no effect on scan paths as these are exactly the features which are already being used to guide those early eye movement responses. Elaborative titles which highlight information beyond these basic aspects of the artworks would only affect those eye movements which occur later in viewing as the focal survey phase comes into play. Given the quick gist extraction of artworks we suggest that rather than giving descriptive or elaborative titles as prior contextual information (titles) a more effective contextual guide for viewing behaviour would be to give simple basic-level or superordinate categorical information. If categorical information can be used to guide expectations of what artworks depict then we would expect to see differences in eye movement control, especially early in the viewing period. We know that briefly presented commonly encountered heterogenous scenes (10-107ms) have been found to be easily categorized at basic-levels (33–43). Initial holistic impressions of artworks are also reported with very brief exposures and show stable ratings with increasing exposure durations based on style, structure, semantic meaning, complexity, harmony, and order (15,21,44–47). There is some evidence that aesthetic judgements of beauty, liking, impressiveness, and specialness can be extracted with brief exposures, but other studies report that this develops over time (21,48,49).”

3. In the method, have you thought of using mixed model analysis? This type is more effective in handling smaller sample sizes and controlling for type I errors by considering random effects and fixed effects together. Can you please explore the possibility of integrating this to your current study.

Response: We appreciate the suggestion and we did explore it but have elected not to take this approach for the following reasons. First, while the sample size might appear small these experiments were well powered. Second, as you state to do this we would include the random effects of participants and artworks into model of our DV’s. Including the participants is straightforward as it would account for effects of individual differences. The inclusion of artworks is more problematic though. In the experiments reported we adopted an approach in which we attempted to minimise (wash out) the effect of artwork by randomly varying the artwork and label association for each participant. Thus, each participant may have seen to same artwork with a different label or the same label as another participant. The aim of this was to introduce non-systematic variation in the data and minimise any systematic effect of artwork in participants viewing behaviour. Given this variation in what participants experienced it is quite odd to include artwork as a random effect into a model and it seems likely to be detrimental to the drawing clear conclusions. However, while an linear mixed model approach didn’t sit particularly well without design the Bayesian analysis suggested by the reviewer did. It is one which we had not considered and would fit well with the general design and thrust of the ANOVA outcomes reported. To this end we reanalysed the data using Bayes Factor repeated measures ANOVAs rather than Linear Mixed Models and have included the outcome of this throughout the manuscript.

4. Reading the discussion, it is a bit difficult to extract the practical implications of this work. I would suggest that you add this as a separate section after the discussion.

Response: Thank you we have now included a section on practical implication as you suggest. This new section is shown below:

Added at the end of the Discussion section:

“Practical Implications

We are hesitant to offer practical implications for artists and those who seek to improve accessibility to art. However, it is notable that art viewers like to have labels and context to artworks which is most obviously shown by the cyclical pattern of gaze: first art - then label - then back to art gaze behaviour recorded in museum visits (8,9). If label information is available most viewers will use it. The research outlined in this article, and the outcomes of the study reported here, suggest that descriptive titles (28-31) and broader categorical information will not affect gaze behaviour as the gist of an artwork is readily extracted in very few fixations. However, elaborative titles and information about an artwork which serve to reduce ambiguity (29) or highlight more thematic and interpretative aspects (26) of it will be used and will impact on gaze patterns and verbal descriptions. This suggests that if you are an artist or curator (for example) and you want your audience to be drawn to the aspects of the work not captured by the visual structure and semantic meaning then tell them about it.”

5. The limitation and conclusion in the discussion can be separated into a separate section.

Response: We were unclear about the limitation mentioned as this is not immediately obvious to us from the text but, as suggested, the conclusion has been placed in a separate section.

Reviewer’s comments to the Author

Reviewer #1: This is the review for manuscript PONE-D-24-31271 entitled: “Viewing of abstract art follows a gist to survey gaze pattern over time regardless of broad categorical titles”. Therein, the authors describe two experiments in which they showed pieces of art to participants while they recorded their eye movements. Prior to the presentation, art works were labelled. The authors assumed that labelling will affect gaze behavior during viewing the artwork. However, this was not the case. The authors argue that the results can be explained by the labels not adding to the processing of the artwork above what would have been expected by a gist to survey viewing behavior and the inherent properties of abstract pieces of art.

Overall, the manuscript is well written, and the analyses are in principal appropriate for answering the research question. The literature review is comprehensive and the explanation of the results plausible. Eye movement recordings and analyses follow the state-of-the-art and are well described. I have some questions and suggestions concerning the analyses. Please, see my major and minor points below.

1) One alternative explanation could be that the labels were ignored by participants. Did the authors check that labels were recognized and processed? For instance, by asking them after viewing about the label of the artwork? If the authors did not check this, I suggest to at least mention this as an alternative explanation for the missing effect of labelling. A stronger manipulation could be to instruct participants if they find the label a piece of art was given suitable.

Response: This was checked in two ways.

a. As mentioned in the Procedure each participant was given an information sheet outlining the study. This contained a description of the categorical labels and the temporal order of each trial (i.e., label then artwork). Their understanding of this procedure was checked prior to consent gathering.

b. One of the benefits of using an Eyelink eyetracker is that there are two displays: one showing the trial to the participants and one showing the eye movements and trials display (optional) to the experimenter. Throughout experimental recording the experimenter watched the participants responses on this display to ensure that the label was fixated, that participants complied with experimental instructions and engages with the artworks, and finally that there was no significant tracker loss or error. This latter point has been added at the end of the procedure section:

“During the experiment, eye movement responses were monitored on a dedicated eye tracker output display computer monitor to ensure that participants engaged with the task and did actually look at both the label and the artwork.”

2) For the frequentist analysis approach, did the authors maker power calculations as to how many participants would be needed to at least find a small or medium effect of the labels? If not, they could calculate the achieved power post hoc and mention it in the Participants section of the manuscript.

Response: A power analysis was carried out prior to running the experiment to determine the sample size as part of the design process and ethics approval process we followed. This was unintentionally omitted from the original submission. With alpha set at .05, power at .8, the assumption that the correlation between the two levels of the repeated measures factors was ���� and a medium effect size of Cohen’s f=.25 total sample size was determined to be 15. An over recruitment strategy was adopted in cases of participant drop out and tracker loss or random factors such as unexpected fire alarms. With this in mind we have added that information to the participant section:

“With alpha set at .05, power at .8, the assumption that the correlation between the two levels of the repeated measures factors was ���� and a medium effect size of Cohen’s f=.25 total sample size was determined to be 15. An over recruitment strategy was adopted in cases of participant drop out and tracker loss or random factors such as unexpected fire alarms.”

3) In frequentist analyses, a null result cannot be interpreted. It would be of interest, if there is convincing evidence against the effect of category label by running a Bayesian ANOVA, for instance with the open source and easy to use software JASP.

Response: Thank you for this suggestion. We have added the outcome from Bayes Factor repeated measures ANOVAs and further Bayesian post-hoc tests from JASP throughout the results section. The approach we have adopted is to report the outcomes each statistical tests as originally reported followed by its corresponding Bayesian analysis outcome.

Please note, during the reanalysis of the data it became apparent that some of the original values for the F-ratios and p values (etc) were incorrectly entered into the original submission. This has been corrected in this revision. It is at all obvious how this error has crept in the manuscript despite very close examination of the original data files. However, it is important to note that none of the findings and conclusions have been changed because of this. All figures were correct in the original manuscript and remain unchanged here. It only affected the numbers in the reports of some of the ANOVAs.

Due to the added analysis and change in some values in the text we have replaced the ANOVA analysis sections of the Results.

Added to the Data Analysis part of the Methods section:

“The experiments were designed under a frequentist approach and consequently data were analysed using Analysis of Variance (ANOVA) with factors of Label and/or Time as detailed in the Results section. To supplement this a series of Bayesian repeated measures ANOVAs are reported with subject and all repeated measures as random slopes for all repeated measures (JASP version 0.19.1; jasp-stats.org). Bayes factors are a way to measure the relative support that data provide to competing models of the data. These were computed using the default prior options for the effects within JASP (i.e., r scale=0.5 for the fixed effects; r scale =1 for the random effects and r scale=0.354 for the covariates). Bayes factors (BF01) are reported here. These express the probability of

---

## [Decision Letter · Decision Letter 1]

Dear Dr. McSorley,

Thank you for submitting your manuscript to PLOS ONE. After careful consideration, we feel that it has merit but does not fully meet PLOS ONE’s publication criteria as it currently stands. Therefore, we invite you to submit a revised version of the manuscript that addresses the points raised during the review process.

We look forward to receiving your revised manuscript.

Kind regards,

Hosam Al-Samarraie

Academic Editor

PLOS ONE

Additional Editor Comments:

Thank you for submitting the revised version back to us. Despite some improvement, the reviewer and I still believe that further improvement is needed.

Please review the reviewer's comments and respond to them accordingly. Here are my comments:

The manuscript would benefit from thorough proofreading, as there are many places lacking commas, correct spelling, etc.

In the discussion when you refer to previous studies you need to indicate the number along with the name of authors. “For example, some works of abstract art are described as more calming and static, such as those by Piet Mondrian or Josef Albers,”

The citation must be checked throughout the manuscript.

I found the assumption here “Thus, eye movements would be expected to be spread more widely than those evoked by static paintings.” Can benefit from more explanation. There are many previous studies that have indicated this too. Perhaps linking to possible individual differences or characteristics might help explain why.

Also the practical implications section should be placed after the discussion. The conclusion section comes at the end.

Reviewers' comments:

Reviewer's Responses to Questions

**Comments to the Author**

Reviewer #1: (No Response)

2. Is the manuscript technically sound, and do the data support the conclusions?

Reviewer #1: Partly

3. Has the statistical analysis been performed appropriately and rigorously?

Reviewer #1: N/A

4. Have the authors made all data underlying the findings in their manuscript fully available?

Reviewer #1: Yes

5. Is the manuscript presented in an intelligible fashion and written in standard English?

Reviewer #1: No

Reviewer #1: This is the review for Revision 1 of the manuscript: “Viewing of abstract art follows a gist to survey gaze pattern over time regardless of broad categorical titles”. The authors could address some of my points. It is a pity that the authors cannot compare the same artwork with different labels, but only report the overall mean differences between viewing of art works and labels. To me, that would have been a more direct test of the hypothesis and could easily be tested in a third experiment. Based on the revised manuscript, I have some more comments. Please, see below.

Design: should the factor Time not be mentioned in the Design section and power analyses? So far it only includes the factor “Label”. But Time is analyzed and theoretically motivated.

I guess the authors used G*Power 3.1. I think, citing this source would be appropriate.

While I appreciate the use of Bayesian inference to draw conclusions on the reported null effects, the results confused me at first glance. The probability of the alternative over the null hypothesis is BF10. The evidence for the Null hypotheses over the alternative is usually labelled BF01. To ease understanding of the BFs, it may make sense to report the BF according to the reported result. Thus, for instance, instead of reporting BF01=1.326x10-23 (p.14 l.298) which is basically 0 evidence in favor of the null, it makes sense to report the BF10, that is evidence for the difference between the second- and third-time interval.

I don’t understand the expression: “Label BF01/Null model BF01=482.206/249.356=9.55” What do the authors mean? Dividing Label B01 by Null model BF01 results in 1.93, not 9.55.

“Methods” is the proper title including Materials. No need to change that to “Materials and Methods”

Minor language errors:

p.5 l.87: «paintings of Jacob and this Twelve Sons” should be “paintings of Jacob and his Twelve Sons

p.11 “all repeated measures as random slopes for all repeated measures (JASP version 0.19.1; jasp-stats.org).” Do the authors mean: “for all dependent measures” ?

**Do you want your identity to be public for this peer review?** For information about this choice, including consent withdrawal, please see our Privacy Policy

Reviewer #1: No

---

## [Author Response · Author response to Decision Letter 2]

9 Jan 2025

Editors Comments:

1. The manuscript would benefit from thorough proofreading, as there are many places lacking commas, correct spelling, etc.

Response: We have done this. Changes have been made throughout the manuscript. We have also changed spelling from UK to US English but certain manners of expression might differ and are difficult for us, as native UK English speakers, to identify.

2. In the discussion when you refer to previous studies you need to indicate the number along with the name of authors. “For example, some works of abstract art are described as more calming and static, such as those by Piet Mondrian or Josef Albers,”

The citation must be checked throughout the manuscript.

Response: this has been fixed and citations have been checked

4. I found the assumption here “Thus, eye movements would be expected to be spread more widely than those evoked by static paintings.” Can benefit from more explanation. There are many previous studies that have indicated this too. Perhaps linking to possible individual differences or characteristics might help explain why.

Response: This sentence: “Thus, eye movements would be expected to be spread more widely than those evoked by static paintings” sits within a wider context and is not at all treated as an assumption. See below for the surrounding text. As can be seen, explanations and evidence (with citations) are provided and described. Are there specific studies we are missing that you have in mind?

“A further suggestion has been made that dynamic paintings elicit a broader distribution of attention due to the lack of a small number of specific areas of interest in the works (87,88). The works are decentralized, polyphonic and polyfocal. Thus, eye movements would be expected to be spread more widely than those evoked by static paintings. Indeed, it has been suggested that for dynamic action paintings, eye movements would be more common as there is nowhere to settle for extended periods and it is this that elicits the perception that the works are vital and dynamic (89,90). The term “polyfocal all-over” has been suggested to describe this pattern of behaviour on elements in works, such as those by Pollock, that are homogenous and have no obvious centre within them (91). Support for this has been reported by (92). They found that the distribution of fixations was much broader for more dynamic paintings compared with those rated as less dynamic although fixation duration was found not to differ. Whereas (79), using abstract works associated with descriptions of static and dynamic, found that those rated more highly for dynamism evoked an increased number of fixations with shorter durations, a greater number of saccades, with faster saccade velocity and increased ratings of pleasantness. Given this, it follows that no effect of categorical label was found here because it is the painting itself which evokes the eye movement response with more dynamism perceived in the works eliciting more dynamism in viewing behaviour. Top-down expectation of content has little or no role to play in evoking more or less “action viewing” behaviour.”

4. Also the practical implications section should be placed after the discussion. The conclusion section comes at the end.

Response: This has been changed

Reviewers' comments:

1. Design: should the factor Time not be mentioned in the Design section and power analyses? So far it only includes the factor “Label”. But Time is analyzed and theoretically motivated.

Response: We took the Design section as a place to outline the experimental design rather than the place that reflects the data analysis plan, so we did not include the time partition of gaze responses there. However, we do appreciate your point, and following your suggestion, we have now included a specific mention of time as a factor in the Design section. We have added the following text: “To examine changes in fixation and eye movement behavior across the viewing period we introduced a second factor of Time. Eye-tracking data was sectioned into that which occurred early (0 to 2 seconds), mid-way (2 to 7 seconds), and towards the end of the trial (7 seconds to trial end).”

Regarding power analysis, we included the TIME factor in the estimation of effect size by modelling in G*Power with groups set at 1 and measurements set at 9. We have included the following in the Participant section

“Sample size was determined using G*Power 3.1 (50). Number of groups was set at 1 and measurements at 9 (3X3 two-way repeated measures design), with Alpha set at .05 and power at .8. The assumption that the correlation between the two levels of the repeated measures factors was set at r=.5 and a medium effect size of Cohen’s f=.25 was selected. As a result, the total sample size was determined to be 15. This is in line with comparable studies examining the impact of art on eye movement behaviour (see studies examining eye movements in changes in response to art discussed in the introduction, which is typically around 20 overall or per condition: 14, 16, 20, 26-30).”

2. I guess the authors used G*Power 3.1. I think, citing this source would be appropriate.

Response: Reference to G*Power has been added along with the reference (see above).

3. While I appreciate the use of Bayesian inference to draw conclusions on the reported null effects, the results confused me at first glance. The probability of the alternative over the null hypothesis is BF10. The evidence for the Null hypotheses over the alternative is usually labelled BF01. To ease understanding of the BFs, it may make sense to report the BF according to the reported result. Thus, for instance, instead of reporting BF01=1.326x10-23 (p.14 l.298) which is basically evidence in favor of the null, it makes sense to report the BF10, that is evidence for the difference between the second- and third-time interval.

Response: We introduced the Bayes analysis to identify if there was convincing evidence for the null model (and thus against an effect of category label) in response to the suggestion in our first round of reviews. From this perspective, it makes more sense to report BF01 rather than BF10. However, we appreciate that within the narrative context of the experiments presented here, BF10 would ease understanding. For this reason, we have adopted the suggestion here and recomputed Bayes RM ANOVAs to reported BF10’s. Amendments have been made throughout the Results section.

4. I don’t understand the expression: “Label BF01/Null model BF01=482.206/249.356=9.55” What do the authors mean? Dividing Label B01 by Null model BF01 results in 1.93, not 9.55.

Response: This was an error on our part and has been fixed.

5. “Methods” is the proper title including Materials. No need to change that to “Materials and Methods”

Response: We followed the suggestions on the Plos One guidance in naming this section Materials and Methods. As suggested, this has been changed to Methods.

6. Minor language errors:

p.5 l.87: «paintings of Jacob and this Twelve Sons” should be “paintings of Jacob and his Twelve Sons

p.11 “all repeated measures as random slopes for all repeated measures (JASP version 0.19.1; jasp-stats.org).” Do the authors mean: “for all dependent measures” ?

Response: these have been corrected

---

## [Decision Letter · Decision Letter 2]

Dear Dr. McSorley,

Thank you for submitting your manuscript to PLOS ONE. After careful consideration, we feel that it has merit but does not fully meet PLOS ONE’s publication criteria as it currently stands. Therefore, we invite you to submit a revised version of the manuscript that addresses the points raised during the review process.

We look forward to receiving your revised manuscript.

Kind regards,

Hosam Al-Samarraie

Academic Editor

PLOS ONE

Journal Requirements:

Additional Editor Comments:

Dear authors, thank you for your patience and for addressing the major comments in the previous round. Please do consider the reviewers' comments when submitting your revised manuscript

Reviewers' comments:

Reviewer's Responses to Questions

**Comments to the Author**

Reviewer #2: All comments have been addressed

Reviewer #3: All comments have been addressed

2. Is the manuscript technically sound, and do the data support the conclusions?

Reviewer #2: Yes

Reviewer #3: Yes

3. Has the statistical analysis been performed appropriately and rigorously?

Reviewer #2: Yes

Reviewer #3: Yes

4. Have the authors made all data underlying the findings in their manuscript fully available?

Reviewer #2: Yes

Reviewer #3: Yes

5. Is the manuscript presented in an intelligible fashion and written in standard English?

Reviewer #2: Yes

Reviewer #3: Yes

Reviewer #2: Your article is well-revised and meets a high standard for publication. I have a minor suggestion: consider including a visualization of raw eye movement data overlaid on the abstract art, showing fixation points and saccades for a representative 10-second segment from both experiments. This would help the audience intuitively grasp the gaze patterns before presenting the bar graphs. Additionally, a brief explanation in the results section would enhance clarity and provide better context for the statistical summaries

Reviewer #3: (No Response)

**Do you want your identity to be public for this peer review?** For information about this choice, including consent withdrawal, please see our Privacy Policy

Reviewer #2: **Yes: ** Suraj Upadhyaya

Reviewer #3: No

---

## [Author Response · Author response to Decision Letter 3]

9 May 2025

Reviewer comments:

1. Your article is well-revised and meets a high standard for publication. I have a minor suggestion: consider including a visualization of raw eye movement data overlaid on the abstract art, showing fixation points and saccades for a representative 10-second segment from both experiments. This would help the audience intuitively grasp the gaze patterns before presenting the bar graphs. Additionally, a brief explanation in the results section would enhance clarity and provide better context for the statistical summaries.

Response:

While we thank the reviewer for their comment and suggestion. We did consider the addition of the visualizations requested to the manuscript and have elected include examples of these as a new figure with supporting text. To provide a context for data analysis and improve clarity these have been added to the manuscript at the beginning of the Results section. As the art works shown to viewers differed across trial then aggregating scan paths across trials would not make sense. Nor would aggregating scan paths across participant. This leaves the option to include single trial example images of fixation points and saccades scan paths for individual participants from each experiment and condition. The result of this is shown below and has been incorporated into the text as follows:

Addition to the main body of the text of Results section:

“To help the reader visualize the eye movements we recorded example plots of recorded scan paths. These are shown in Fig 2 overlaid upon the abstract artwork that evoked that scan path response. Each artwork and scan path shown is a single trial from different participants. Those on the left are from Experiment 1 (labelled for each participant as: “Landscape”, “Portrait” and “Abstract” from top to bottom) and those on the right are from Experiment 2 (labelled “Still Life”, “Action” and “Abstract” from top to bottom). Plots show fixation locations as blue circles along with the start and end points of each saccade (with a joining straight yellow line). Fixation durations are represented by circle diameter with wider ones indicating longer durations as shown by the number next to each circle which shows the actual duration in milliseconds. The location of the saccade in the scan path sequence is given the yellow number towards the end of each line e.g., 11 would mean the 11th saccade in that scan path sequence for that trial.”

Additional Figure legend in text of manuscript:

“Fig 2: shows example scan paths for six abstract art works for six different participants. The three on the left are from Experiment 1 (labelled for each participant as: “Landscape”, “Portrait” and “Abstract” from top to bottom) and those on the right are from Experiment 2 (labelled “Still Life”, “Action” and “Abstract” from top to bottom). Fixations are plotted as pale blue circles, the center of each is the fixation location while their diameter represents their duration which is also given by the number (in milliseconds) next to each circle. Saccades are shown as straight yellow lines between their start and end points with the yellow number indicating the point at which each saccade as executed in the path sequence (i.e., 1 is the first saccade, 2 is the second and so on).”

The Figure itself is shown in the attachments to the paper submission.

Reviewer's report:

In general, I believe the authors have addressed the issues raised by the reviewers. However, there is one further important point that I suggest to address.

The manuscript would benefit from a more in-depth discussion of the potential impact of individual differences in eye movement patterns while viewing artworks.

Recent research has highlighted this aspect. For example, Zangrossi et al. (2021) demonstrated that individual differences in viewing style (e.g., static vs. dynamic viewers) are stable across different image categories and even persist when participants are watching a blank screen. Specifically, static viewers exhibit fewer but longer fixations, whereas dynamic viewers display a more ambient viewing pattern characterized by a greater number of shorter fixations. A subsequent study by the same group (Celli et al., 2022) found that these viewing styles are associated with stable differences in resting-state brain activity.

Additionally, several studies have reported idiosyncratic eye movement profiles when individuals perform different oculomotor tasks (Poynter et al., 2013), including distinctive "oculomotor signatures" (Bargary et al., 2017; Kennedy et al., 2017).

Taken together, these findings suggest that, at least in part, the results presented in the manuscript may be influenced by interindividual differences, which should be considered in the discussion. This point is particularly critical for the present study since abstract artworks potentially leave more space to interindividual differences (i.e., endogenous processes guiding eye movements) as compared to more structured paintings (e.g., portraits, complex scenes, ‚Ä¶).

References

Bargary, G., Bosten, J. M., Goodbourn, P. T., Lawrance-Owen, A. J., Hogg, R. E., & Mollon, J. D. (2017). Individual differences in human eye movements: An oculomotor signature? Vision Research, 141, 157-169. https://doi.org/10.1016/j.visres.2017.03.001

Celli, M., Mazzonetto, I., Zangrossi, A., Bertoldo, A., Cona, G., & Corbetta, M. (2022). One-year-later spontaneous EEG features predict visual exploratory human phenotypes. Communications Biology, 5(1), 1361. https://doi.org/10.1038/s42003-022-04294-9

Kennedy, D. P., D'Onofrio, B. M., Quinn, P. D., B√∂lte, S., Lichtenstein, P., & Falck-Ytter, T. (2017). Genetic Influence on Eye Movements to Complex Scenes at Short Timescales. Current Biology, 27(22), 3554-3560.e3. https://doi.org/10.1016/j.cub.2017.10.007

Poynter, W., Barber, M., Inman, J., & Wiggins, C. (2013). Individuals exhibit idiosyncratic eye-movement behavior profiles across tasks. Vision Research, 89, 32-38. https://doi.org/10.1016/j.visres.2013.07.002

Zangrossi, A., Cona, G., Celli, M., Zorzi, M., & Corbetta, M. (2021). Visual exploration dynamics are low-dimensional and driven by intrinsic factors. Communications Biology, 4(1), 1100. https://doi.org/10.1038/s42003-021-02608-x

Response:

Many thanks for the suggestion and for the additional references. As a result of reading these the additional text has been added to the Discussion (all additional numbered references have been added to the reference list):

A further consideration in the control of gaze when viewing artworks is a potential role of stable individual differences in viewing behavior and the shift from the Gist-Ambient to the Survey-Focal phase of art viewing. Measures of oculomotor behavior exhibit stable characteristics that can be captured with common descriptive statistics, but they also reveal a wide range in individual responses. This can be seen in large population studies of saccades, anti-saccades and smooth pursuit (1–4). Despite these dissimilarities, recent research has identified stable gaze patterns within individuals. Across a range of diverse tasks individuals produce similar eye movement behaviours such as fixation numbers, fixation durations, and saccade amplitudes (5–10), with a similar consistency also shown in measures of micro-saccadic eye movement behavior such as fixation extent, micro-saccade number and amplitude (11). Recently Bargary et al (12) reported reliable individual pro- and anti- saccade and smooth pursuit eye movement patterns across task and test-retest sessions with eye movement parameters that correlated well within eye movement type. These have been likened to an oculomotor signature (12) and have been related to differences in personality traits (13) and genetic influences (14). Poynter et al (11) identified strong correlations across six eye movement metrics (e.g., fixation rate, duration and size, saccade amplitude, microsaccade rate and amplitude) that were captured by a single underlying latent factor. Zangrossi et al (15) also reported that viewing behavior dynamics of a large sample of participants in response to a wide range of real-world natural scenes was explained well by very few latent variables with only three components explaining about 60% of eye movement dynamics (those that more reflect how or when people look: fixation duration, and number, gaze step direction and amplitude of saccade). Unlike gaze dynamics analysis of the spatial distribution of fixations (where people look: density map, semantic map and saliency map) show a dependency of gaze distribution on image meaning and points of saliency. The low dimensionality of gaze dynamics led Zangrossi et al (15) to identify individual viewing styles into the two stable types of static and dynamic viewers, that reflective automatic intrinsic, endogenous preferences that are executed relatively independently of differences in the external visual environment and remain stable when participants view a blank screen. These viewing styles have been linked to persistent differences in resting-state EEG brain activity (16) shown in static and dynamic viewers tested one year after behavioral experiments reported by Zangrossi et al (15). These differences in resting-state have been linked to higher cortical inhibition and focus on internal processing for static viewers while dynamic viewers have a profile more biased toward cortical excitation and external processing. This interpretation is bolstered by behavioral outcome measures with static viewers showing slightly stronger visual working memory and dynamic viewers showing weaker inhibition of salient but non-relevant stimuli. These stable individual differences in eye movement control and viewing behaviours are likely to also play a role in gaze patterns while viewing artworks and in the eye movement outcomes reported here. This is especially likely when the stimuli viewed were abstract artworks which may encourage eye movements that reflect an individual’s idiosyncratic endogenous internally driven viewing style rather than being influenced by the artwork’s semantic content, something that would be found in more structured paintings such as portraits or landscapes. This is something that would be fruitful to more thoroughly explore in future studies in this area.

---

## [Editor Report · Decision Letter 3]

Viewing of abstract art follows a gist to survey gaze pattern over time regardless of broad categorical titles

PONE-D-24-31271R3

Dear Dr. McSorley,

We’re pleased to inform you that your manuscript has been judged scientifically suitable for publication and will be formally accepted for publication once it meets all outstanding technical requirements.

Kind regards,

Hosam Al-Samarraie

Academic Editor

PLOS ONE

Additional Editor Comments (optional):

Thank you for addressing all the comments from the second round.
---

## [Editor Report · Acceptance letter]

PONE-D-24-31271R3

PLOS ONE

Dear Dr. McSorley,

I'm pleased to inform you that your manuscript has been deemed suitable for publication in PLOS ONE. Congratulations! Your manuscript is now being handed over to our production team.

Kind regards,

on behalf of

Dr Hosam Al-Samarraie

Academic Editor

PLOS ONE